# ADVERSARIAL RETRIEVER-RANKER FOR DENSE TEXT RETRIEVAL

**Hang Zhang[1],**[*] **Yeyun Gong[2],**[†] **Yelong Shen[3], Jiancheng Lv[1], Nan Duan[2], Weizhu Chen[3]**
[1]College of Computer Science, Sichuan University,
[2]Microsoft Research Asia, [3]Microsoft Azure AI
`hangzhang_scu@foxmail.com,`{`yegong,yelong.shen`}`@microsoft.com,`
`lvjiancheng@scu.edu.cn,` {`nanduan,wzchen`}`@microsoft.com`

## ABSTRACT

Current dense text retrieval models face two typical challenges. First, they adopt a siamese dual-encoder architecture to encode queries and documents independently for fast indexing and searching, while neglecting the finer-grained term-wise interactions. This results in a sub-optimal recall performance. Second, their model training highly relies on a negative sampling technique to build up the negative documents in their contrastive losses. To address these challenges, we present *Adversarial Retriever-Ranker* (AR2), which consists of a dual-encoder retriever plus a cross-encoder ranker. The two models are jointly optimized according to a minimax adversarial objective: the retriever learns to retrieve negative documents to cheat the ranker, while the ranker learns to rank a collection of candidates including both the ground-truth and the retrieved ones, as well as providing progressive direct feedback to the dual-encoder retriever. Through this adversarial game, the retriever gradually produces harder negative documents to train a better ranker, whereas the cross-encoder ranker provides progressive feedback to improve retriever. We evaluate AR2 on three benchmarks. Experimental results show that AR2 consistently and significantly outperforms existing dense retriever methods and achieves new state-of-the-art results on all of them. This includes the improvements on Natural Questions R@5 to 77.9% (+2.1%), TriviaQA R@5 to 78.2% (+1.4%), and MS-MARCO MRR@10 to 39.5% (+1.3%). Code and models are available at `https://github.com/microsoft/AR2`.

## 1 INTRODUCTION

Dense text retrieval (Lee et al., 2019; Karpukhin et al., 2020) has achieved great successes in a wide variety of both research and industrial areas, such as search engines (Brickley et al., 2019; Shen et al., 2014), recommendation system (Hu et al., 2020), open-domain question answering (Guo et al., 2018; Liu et al., 2020), etc. A typical dense retrieval model adopts a dual-encoder (Huang et al., 2013) architecture to encode queries and documents into low-dimensional embedding vectors, with the relevance between query and document being measured by the similarity between embeddings. In real-world dense text retrieval applications, it pre-computes all the embedding vectors of documents in the corpus, and leverages the approximate nearest neighbor (ANN) (Johnson et al., 2019) technique for efficiency. To train a dense retriever, contrastive loss with negative samples is widely applied in the literature (Xiong et al., 2021; Karpukhin et al., 2020). During training, the model utilizes a negative sampling method to obtain negative documents for a given query-document pair, and then minimizes the contrastive loss which relies on both the positive document and the sampled negative ones (Shen et al., 2014; Chen et al., 2017; Radford et al., 2021).

Recent studies on contrastive learning (Xiong et al., 2021; Karpukhin et al., 2020) show that the iterative "hard-negative" sampling technique can significantly improve the performance compared with "random-negative" sampling approach, as it can pick more representative negative samples to

---

[*]Work is done during internship at Microsoft Research Asia.
[†]Corresponding author

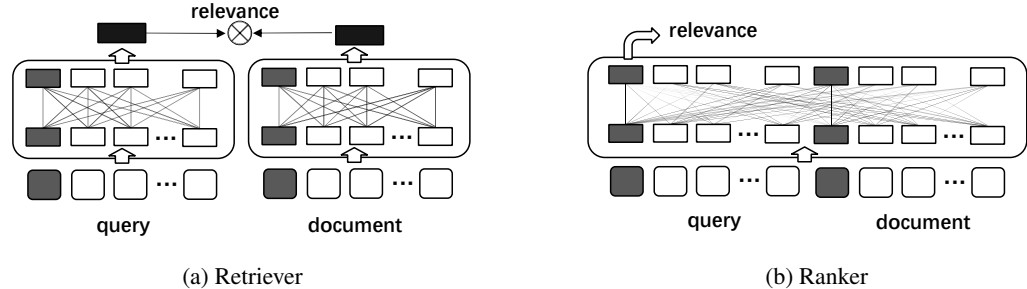

|     |     |
| :-: | :-: |
| (a) Retriever | (b) Ranker |

Figure 1: Illustration of two modules in AR2. (a) Retriever: query and document are encoded independently by a dual-encoder. (b) Ranker: concatenated, jointly encoded by a cross-encoder.

learn a more discriminative retriever. In the work (Qu et al., 2021), it suggests leveraging cross-encoder model to heuristically filter "hard-negative" samples to further improve performance and shows the importance of sampling technique in the contrastive learning.

On the other hand, the model architecture of dual-encoders enables the encoding of queries and documents independently which is essential for document indexing and fast retrieval. However, this ignores the modeling of finer-grained interactions between queries and documents which could be a sub-optimal solution in terms of retrieval accuracy.

Motivated by these phenomena, we propose an *Adversarial **R**etriever-**R**anker* (AR2) framework. The intuitive idea of AR2 is inspired by the "retriever-ranker" architecture in the classical information retrieval systems. AR2 consists of two modules: a dual-encoder model served as the retrieval module in Figure 1a and a cross-encoder model served as the ranker module in Figure 1b. The cross-encoder model takes the concatenation of a query and document as input text, and can generate more accurate relevance scores compared with the dual-encoder model, since it can fully explore the interactions between the query and document through a self-attention mechanism using a conventional transformer model (Vaswani et al., 2017; Guo et al., 2020). Instead of training "retriever-ranker" modules independently in some IR systems (Manning et al., 2008; Mitra & Craswell, 2017), AR2 constructs a unified minimax game for training the retriever and ranker models interactively, as shown in Figure 2.

In particular, AR2 adopts a minimax objective from the adversarial game (Goodfellow et al., 2014) where the retrieval model is optimized to produce relevant documents to fool the ranker model, whereas the ranker model learns to distinguish the ground-truth relevant document and retrieved ones by its opponent retrieval model. Within the adversarial "retriever-ranker" training framework, the retrieval model receives the smooth training signals from the ranker model which helps alleviate the harmful effects of "false-negative" issues. For example, a "false-negative" example which is rated as high-relevance by the ranker model, will also be granted with high probability by retrieval model in order to fool the ranker, meanwhile the ranker model with better generalization capability is more resistant to label noises compared to the retrieval model.

In the empirical studies of AR2, we further introduce a distillation regularization approach to help stabilize/improve the training of the retriever. Intuitively, the retriever would converge to sharp conditional-probabilities over documents given a query within the adversarial training framework, i.e., high retrieval probabilities for the top relevant documents and near-zero retrieval ones for the rest. However, it is not a desirable

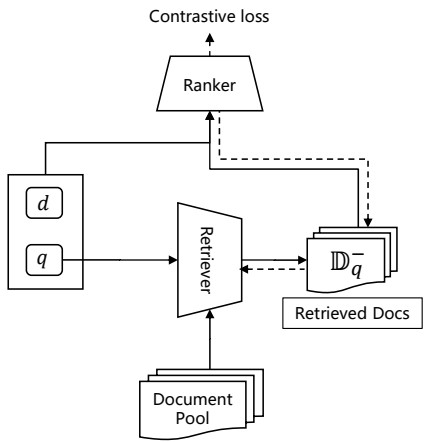

Figure 2: Illustration of the AR2 training pipeline. $q$, $d$, and $\mathbb{D}_q^-$ represent the query, positive document, and retrieved documents, respectively.

property as it might impede exploring dif-

ferent documents during training. Thus, we incorporate the distillation loss between the retriever and ranker models as a smooth term for further improvement.

In experiments, we evaluate AR2 on three widely used benchmarks for dense text retrieval: Natural Questions, Trivia QA and MS-MARCO. Experimental results show that AR2 achieves state-of-the-art performance on all these datasets. Meanwhile, we provide a comprehensive ablation study to demonstrate the advantage of different AR2 components.

## 2 PRELIMINARIES

**Dense Text Retrieval:** We mainly consider a contrastive-learning paradigm for dense text retrieval in this work, where the training set consists of a collection of text pairs. $C = \{(q_1, d_1), ..., (q_n, d_n)\}$. In the scenario of open-domain question answering, a text pair $(q, d)$ refers to a question and a corresponding document which contains the answer. A typical dense retrieval model adopts a dual encoder architecture, where questions and documents are represented as dense vectors separately and the relevance score $s_\theta(q, d)$ between them is measured by the similarity between their embeddings:

$$s_\theta(q, d) = \langle E(q; \theta), E(d; \theta)) \rangle \tag{1}$$

where $E(\cdot; \theta)$ denotes the encoder module parameterized with $\theta$, and $\langle \cdot \rangle$ is the similarity function, e.g., inner-product, Euclidean distance. Based on the embeddings, existing solutions generally leverage on-the-shelf fast ANN-search (Johnson et al., 2019) for efficiency.

A conventional contrastive-learning algorithm could be applied for training the dual encoders (Shen et al., 2014; Chen et al., 2017; Liu et al., 2020). For example, given a training instance $(q, d)$, we select $n$ negative irrelevant documents $(d_1^-, ..., d_n^-)$ (denoted as $\mathbb{D}_q^-$) to optimize the loss function of the negative log likelihood of the positive document:

$$L_\theta(q, d, \mathbb{D}_q^-) = -\log \frac{e^{\tau s_\theta(q,d)}}{e^{\tau s_\theta(q,d)} + \sum_{i=1}^n e^{\tau s_\theta(q,d_i^-)}} \tag{2}$$

where $\tau$ is a hyper-parameter to control the temperature. Previous works (Shen et al., 2014; Chen et al., 2017; Liu et al., 2020) present an effective strategy on negative document sampling, called "In-Batch Negatives" where negative documents are randomly sampled from a collection of documents which are within the same mini-batch as question-document training pairs.

Recently, some studies e.g., ANCE (Xiong et al., 2021) and Condenser (Gao & Callan, 2021b), have shown that selecting "hard-negatives" in the training can significantly improve the retrieval performance in open-domain question answering. For example, instead of sampling negative document randomly, "hard-negatives" are iteratively retrieved through previous checkpoints of the dual encoder model. However, a more recent work RocketQA (Qu et al., 2021) continues to point out that the retrieved "hard-negatives" could potential be "false-negatives" in some cases, which might limit the performance.

**Generative Adversarial Network:** GANs have been widely studied for generating the realistic-looking images in computation vision (Goodfellow et al., 2014; Brock et al., 2018). In the past few years, the idea of GANs has been applied in information retrieval (Wang et al., 2017). For example, IRGAN (Wang et al., 2017), proposes a minimax retrieval framework which constructs two types of IR models: a generative retrieval model and a discriminative retrieval model. The two IR models are optimized through a minimax game: the generative retrieval model generates relevant documents that look like ground-truth relevant documents to fool the discriminative retrieval model, whereas the discriminative retrieval model learns to draw a clear distinction between the ground-truth relevant documents and the generated ones made by its opponent generative retrieval model. The minimax objective is formulated as:

$$J^{G^*, D^*} = \min_\theta \max_\phi E_{d \sim p_{\text{true}}(\cdot|q)} \left[\log D_\phi(d, q)\right] + E_{d^- \sim G_\theta(\cdot|q)} \left[\log\left(1 - D_\phi(d^-, q)\right)\right] \tag{3}$$

where $G_\theta(\cdot|q)$ and $D_\phi(d^-, q)$ denote the generative retrieval model and discriminative retrieval model in IRGAN, respectively. It is worth noting the original IRGAN model doesn't work for dense retrieval tasks as it doesn't contain the dual-encoder model for document indexing or fast retrieval.

## 3 METHOD

In this section, we introduce the proposed adversarial retriever-ranker (**AR2**) approach. It consists of two modules: the dual-encoder retriever module $G_\theta$ as in Figure 1a, and the cross-encoder ranker module $D_\phi$ as in Figure 1b. $G_\theta$ and $D_\phi$ computes the relevance score between question and document as follows:

$$G_\theta(q, d) = E_\theta(q)^T E_\theta(d)$$
$$D_\phi(q, d) = \mathbf{w}_\phi{}^T E_\phi([q, d])$$

(4)

where $E_\theta(\cdot)$ and $E_\phi(\cdot)$ are language model encoders which can be initialized with any pre-trained language model, $\mathbf{w}_\phi$ is the linear projector in $D_\phi$, and $[q, d]$ is the concatenation of question and document.

In **AR2**, the retriever and ranker modules are optimized jointly through a contrastive minimax objective:

$$J^{G^*, D^*} = \min_\theta \max_\phi \mathbf{E}_{\mathbb{D}_q^- \sim G_\theta(q, \cdot)} [\log p_\phi(d|q; \mathbb{D}_q)]$$

(5)

where $\mathbb{D}_q^-: \{d_i^-\}_{i=1}^n$ is the set of $n$ negative documents sampled by $G_\theta(q, \cdot)$ given $q$, and $p_\phi(d|q; \mathbb{D}_q)$ denotes the probability of selecting the ground-truth document $d$ from the document set $\mathbb{D}_q$ ($\mathbb{D}_q = \{d\} \cup \mathbb{D}_q^-$) by the ranker module $D_\phi$;

$$p_\phi(d|q; \mathbb{D}_q) = \frac{e^{\tau D_\phi(q, d)}}{\sum_{d' \in \mathbb{D}_q} e^{\tau D_\phi(q, d')}}$$

(6)

According to the objective function (Eqn. 5), the dual-encoder retrieval model $G_\theta(q, \cdot)$ would try to sample the high-relevant documents to fool the ranker model, whereas the ranker model $D_\phi(q, \cdot)$ is optimized to draw distinctions between ground-truth passage and the ones sampled by $G_\theta(q, \cdot)$. We present the illustration of the AR2 framework in Figure 2. In order to optimize the minimax objective function, we adopt a conventional iterative-learning mechanism to optimize the retriever and ranker modules coordinately.

### 3.1 TRAINING THE RANKER $D_\phi$

Given the fixed retriever $G_\theta$, the ranker model $D_\phi$ is updated by maximizing the log likelihood of selecting ground-truth $d$ from set $\mathbb{D}_q$ given a query $q$:

$$\phi^* = \text{argmax}_\phi \log p_\phi(d|q; \mathbb{D}_q)$$

(7)

where $\mathbb{D}_q$ consists of ground-truth document $d$ and negative document set $\mathbb{D}_q^-$. $\mathbb{D}_q^-$ is sampled by $G_\theta$ according to Eqn. 5. In experiments, we first retrieve top-100 negative documents, and then randomly sample $n$ examples from them to obtain $\mathbb{D}_q^-$.

### 3.2 TRAINING RETRIEVER $G_\theta$

With fixing the ranker $D_\phi$, the model parameters $\theta^*$ for the retriever $G_\theta$ is optimized by minimizing the expectation of log likelihood of function. In particular, by isolating $\theta$ from the minimax function (Eqn. 5), the objective for the retriever can be written as:

$$\theta^* = \text{argmin}_\theta J^\theta = \mathbf{E}_{\mathbb{D}_q^- \sim G_\theta(q, \cdot)} [\log p_\phi(d|q; \mathbb{D}_q)]$$

(8)

However, it is intractable to optimize $\theta$ directly through Eqn. 8, as the computation of probability $\mathbb{D}_q^- \sim G_\theta(q, \cdot)$ does not follow a close form. Thus, we seek to minimize an alternative upper-bound of the loss criteria:

$$J^\theta \leq \hat{J}^\theta = \mathbf{E}_{d^- \sim p_\theta(\cdot|q; \mathbb{D}_q^-)} [\log p_\phi(d|q; \{d, d^-\})]$$

(9)

The detailed deviation of Eqn. 9 is provided in the Appendix A.1. Therefore, the gradient of parameter $\theta$ can be computed as the derivative of $\hat{J}^\theta$ with respect to $\theta$:

$$\nabla_\theta \hat{J}^\theta = \mathbf{E}_{d^- \sim p_\theta(\cdot|q; \mathbb{D}_q^-)} \nabla_\theta \log p_\theta(d^-|q; \mathbb{D}_q^-) [\log p_\phi(d|q; \{d, d^-\})]$$

(10)

---

**Algorithm 1** Adversarial Retriever-Ranker (AR2)

---

**Require:** Retriever $G_\theta$; Ranker $D_\phi$; Document pool $\mathbb{D}$; Training dataset $C$.
 1: Initialize the retriever $G_\theta$ and the ranker $D_\phi$ with pre-trained language models.
 2: Train the warm-up retriever $G_\theta^0$ on training dataset $C$.
 3: Build ANN index on $\mathbb{D}$
 4: Retrieve negative samples on $\mathbb{D}$.
 5: Train the warm-up ranker $D_\theta^0$
 6: **while** AR2 has not converged **do**
 7:    **for** Retriever training step **do**
 8:       Sample $n$ documents $\{d_i^-\}_n$ from ANN index.
 9:       Update parameters of the retriever $G_\theta$.
10:    **end for**
11:    Refresh ANN Index.
12:    **for** Ranker training step **do**
13:       Sample $n$ hard negatives $\{d_i^-\}_n$ from ANN index.
14:       Update parameters of the ranker $D_\phi$.
15:    **end for**
16: **end while**

---

Here, the same approach is applied to obtain set $\mathbb{D}_q^-$ as in Eqn. 7.

**Regularization:** we further introduce a distillation regularization term in $G_\theta$'s training, which encourages the retriever model to distill from the ranker model.

$$J_\mathcal{R}^\theta = H(p_\phi(\cdot|q; \mathbb{D}), p_\theta(\cdot|q; \mathbb{D})) \tag{11}$$

$H(\cdot)$ is the cross entropy function. $p_\phi(\cdot|q; \mathbb{D})$ and $p_\theta(\cdot|q; \mathbb{D})$ denote the conditional probabilities of document in the whole corpus $\mathbb{D}$ by the ranker and the retriever model, respectively. In practice, we also limit the sampling space over documents to a fixed set, i.e., $\mathbb{D}_q = \{d\} \cup \mathbb{D}_q^-$. Thus the regularization loss for the retriever model can be rewritten as:

$$J_\mathcal{R}^\theta = H\left(p_\phi(\cdot|q; \mathbb{D}_q), p_\theta(\cdot|q; \mathbb{D}_q)\right) \tag{12}$$

### 3.3 INDEX REFRESH

During each training iteration of retriever and ranker models in AR2, we refresh the document index to keep the retrieved document set updated. To build the document index, we take the document encoder from the retrieval model to compute the embeddings $E(d; \theta)$ for every document $d$ from the corpus: $d \in C$, and then build the inner-product based ANN search index with FAISS tool.

In summary, Algorithm 1 shows the full implementation details of the proposed AR2.

## 4 EXPERIMENTS

### 4.1 DATASETS

We conduct experiments on three popular benchmarks: Natural Questions (Kwiatkowski et al., 2019), Trivia QA (Joshi et al., 2017), and MS-MARCO Passage Ranking (Nguyen et al., 2016).

Natural Questions (NQ) collects real questions from the Google search engine and each question is paired with an answer span and golden passages from the Wikipedia pages. In NQ, the goal of the retrieval stage is to find positive passages from a large passage pool. We report Recall of top-$k$ (R@k), which represents the proportion of top k retrieved passages that contain the answers.

Trivia QA is a reading comprehension corpus authored by trivia enthusiasts. Each sample is a $\langle question, answer, evidence \rangle$ triple. In the retrieval stage, the goal is to find passages that contain the answer. We also use Recall of top-$k$ as the evaluation metric for Trivia QA.

MS-MARCO Passage Ranking is widely used in information retrieval. It collects real questions from the Bing search engine. Each question is paired with several web documents. Following previous

Table 1: The comparison of the first-stage retrieval performance on Natural Questions test set, Trivia QA test set, and MS-MARCO dev set. The results of the first two blocks are from published papers. If the results are not provided, we mark them as "-".

| | Natural Questions | | | Trivia QA | | | MS-MARCO | | |
|---|---|---|---|---|---|---|---|---|---|
| | R@5 | R@20 | R@100 | R@5 | R@20 | R@100 | MRR@10 | R@50 | R@1k |
| BM25 (Yang et al., 2017) | - | 59.1 | 73.7 | - | 66.9 | 76.7 | 18.7 | 59.2 | 85.7 |
| GAR (Mao et al., 2021a) | 60.9 | 74.4 | 85.3 | 73.1 | 80.4 | 85.7 | - | - | - |
| doc2query (Nogueira et al., 2019b) | - | - | - | - | - | - | 21.5 | 64.4 | 89.1 |
| DeepCT (Dai & Callan, 2019) | - | - | - | - | - | - | 24.3 | 69.0 | 91.0 |
| docTTTTTquery (Nogueira et al., 2019a) | - | - | - | - | - | - | 27.7 | 75.6 | 94.7 |
| DPR (Karpukhin et al., 2020) | - | 78.4 | 85.3 | - | 79.3 | 84.9 | - | - | - |
| ANCE (Xiong et al., 2021) | - | 81.9 | 87.5 | - | 80.3 | 85.3 | 33.0 | - | 95.9 |
| RDR (Yang & Seo, 2020) | - | 82.8 | 88.2 | - | 82.5 | 87.3 | - | - | - |
| ColBERT (Khattab & Zaharia, 2020) | - | - | - | - | - | - | 36.0 | 82.9 | 96.8 |
| RocketQA (Qu et al., 2021) | 74.0 | 82.7 | 88.5 | - | - | - | 37.0 | 85.5 | 97.9 |
| COIL (Gao et al., 2021) | - | - | - | - | - | - | 35.5 | - | 96.3 |
| ME-BERT (Luan et al., 2021) | - | - | - | - | - | - | 33.8 | - | - |
| Joint Top-k (Sachan et al., 2021a) | 72.1 | 81.8 | 87.8 | 74.1 | 81.3 | 86.3 | - | - | - |
| Individual Top-k (Sachan et al., 2021a) | 75.0 | 84.0 | 89.2 | 76.8 | 83.1 | 87.0 | - | - | - |
| PAIR (Ren et al., 2021) | 74.9 | 83.5 | 89.1 | - | - | - | 37.9 | 86.4 | 98.2 |
| DPR-PAQ (Oğuz et al., 2021) | | | | | | | | | |
| -BERT$_{base}$ | 74.5 | 83.7 | 88.6 | - | - | - | 31.4 | - | - |
| -RoBERTa$_{base}$ | 74.2 | 84.0 | 89.2 | - | - | - | 31.1 | - | - |
| Condenser (Gao & Callan, 2021b) | - | 83.2 | 88.4 | - | 81.9 | 86.2 | 36.6 | - | 97.4 |
| coCondenser (Gao & Callan, 2021a) | 75.8 | 84.3 | 89.0 | 76.8 | 83.2 | 87.3 | 38.2 | - | 98.4 |
| AR2-$G^0$ | 69.7 | 80.8 | 87.1 | 74.4 | 81.7 | 86.6 | 34.8 | 84.2 | 98.0 |
| AR2-$G$ | **77.9** | **86.0** | **90.1** | **78.2** | **84.4** | **87.9** | **39.5** | **87.8** | **98.6** |

works (Ren et al., 2021; Qu et al., 2021), we report MRR@10, R@50, R@1k on the dev set. Mean Reciprocal Rank (MRR) is the mean of Reciprocal Rank(RR) across questions, calculated as the reciprocal of the rank where the first relevant document was retrieved.

## 4.2 IMPLEMENTATION DETAILS

First step, we follow the experiments in Sachan et al. (2021b) and Gao & Callan (2021a) to continuous pre-training the ERNIE-2.0-base model (Sun et al., 2020) with Inverse Cloze Task (ICT) training (Lee et al., 2019) for NQ and TriviaQA datasets, and coCondenser training (Gao & Callan, 2021a) for MS-MARCO dataset.

Second step, we follow the experiment settings of DPR (Karpukhin et al., 2020) to train a warm-up dual-encoder retrieval model $\mathbf{G^0}$. It is initialized with the continuous pretrained ERNIE-2.0-based model we obtained in step one. Then we train a warm-up cross-encoder model $\mathbf{D^0}$ initialized with the ERNIE-2.0-Large. $\mathbf{D^0}$ learns to rank the Top-k documents selected by $\mathbf{G^0}$ with contrastive learning. The detailed hyper-parameters in training are listed in Appendix A.3.

Third step, we iteratively train the ranker (AR2-$\mathbf{D}$) model initialized with ERNIE-2.0-large and the retriever (AR2-$\mathbf{G}$) initialized with $\mathbf{G}^0$ according to Algorithm 1. The number of training iterations is set to 10. During each iteration of training, the retriever model is scheduled to train with 1500 mini-batches, while the ranker model is scheduled to train with 500 mini-batches. The document index is refreshed after each iteration of training. The other hyper-parameters are shown in Appendix A.3.

All the experiments in this work run on 8 NVIDIA Tesla A100 GPUs. The implementation code of AR2 is based on Huggingface Transformers (Wolf et al., 2020) utilizing gradient checkpointing (Chen et al., 2016), Apex[1], and gradient accumulation to reduce GPU memory consumption.

## 4.3 RESULTS

**Performance of Retriever AR2-G:** The comparison of retrieval performance on NQ, Trivia QA, and MS-MARCO are presented in Table 1.

We compare AR2-$\mathbf{G}$ with previous state-of-the-art methods, including sparse and dense retrieval models. The top block shows the performance of sparse retrieval methods. BM25 (Yang et al., 2017) is a traditional sparse retriever based on the exact term matching. DeepCT (Dai & Callan, 2019) uses

---

[1]https://github.com/NVIDIA/apex

Table 2: Performance of rankers before and after AR2 training on NQ test set.

| Retriever | Ranker | R@1 | R@5 | R@10 |
|---|---|---|---|---|
| AR2-$G^0$ | - | 48.3 | 69.7 | 76.2 |
| | AR2-$D^0$ | 60.6 | 78.7 | 82.6 |
| | AR2-D | 64.2 | 79.0 | 82.6 |
| AR2-G | - | 58.7 | 77.9 | 82.5 |
| | AR2-$D^0$ | 61.1 | 80.1 | 84.3 |
| | AR2-D | 65.6 | 81.5 | 84.9 |

Table 3: Performance of AR2-G on NQ test set with different negative sample size $n$.

| | R@1 | R@5 | R@20 | R@100 | Latency |
|---|---|---|---|---|---|
| $n=1$ | 56.3 | 76.4 | 85.3 | 89.7 | 210ms |
| $n=5$ | 57.8 | 76.9 | 85.3 | 89.7 | 330ms |
| $n=7$ | 58.0 | 77.2 | 85.2 | 89.7 | 396ms |
| $n=11$ | 58.0 | 77.1 | 85.4 | 89.8 | 510ms |
| $n=15$ | 57.8 | 77.3 | 85.6 | 90.1 | 630ms |

Table 4: Comparison of AR2 and IRGAN.

| | R@1 | R@5 | R@20 | R@100 |
|---|---|---|---|---|
| AR2 | 58.7 | 77.9 | 86.0 | 90.1 |
| IRGAN | 55.2 | 75.2 | 84.5 | 89.2 |

Table 5: Effect of regularization in AR2.

| | R@1 | R@5 | R@20 | R@100 | Entropy |
|---|---|---|---|---|---|
| AR2-G | 58.7 | 77.9 | 86.0 | 90.1 | 2.10 |
| – w/o $R$ | 57.8 | 77.3 | 85.6 | 90.1 | 1.70 |

BERT to dynamically generate lexical weights to augment BM25 Systems. doc2Query (Nogueira et al., 2019b), docTTTTTQuery (Nogueira et al., 2019a), and GAR (Mao et al., 2021a) use text generation to expand queries or documents to make better use of BM25. The middle block lists the results of strong dense retrieval methods, including DPR (Karpukhin et al., 2020), ANCE (Xiong et al., 2021), RDR (Yang & Seo, 2020), RocketQA (Qu et al., 2021), Joint and Individual Top-k (Sachan et al., 2021a), PAIR (Ren et al., 2021), DPR-PAQ (Oğuz et al., 2021), Condenser (Gao & Callan, 2021b). coCondenser (Gao & Callan, 2021a), ME-BERT (Luan et al., 2021), CoIL (Gao et al., 2021). These methods improve the performance of dense retrieval by constructing hard negative samples, jointly training the retriever and downstream tasks, pre-training, knowledge distillation, and multi-vector representations.

The bottom block in Table 1 shows the results of proposed AR2 models. AR2-$G^0$ refers to the warm-up retrieval model in AR2 (details can be found in section 4.2) which leverages the existing continuous pre-training technique for dense text retrieval tasks. i.e., it shows a better performance compared with DPR (Karpukhin et al., 2020) and ANCE (Xiong et al., 2021), etc approaches that do not adopt the continuous pre-training procedure. We also observed that AR2-G: the retrieval model trained with the adversary framework, significantly outperforms the warm-up AR2-$G^0$ model, and achieves new state-of-the-art performance on all three datasets.

## 4.4 Analysis

In this section, we conduct a set of detailed experiments on analyzing the proposed AR2 training framework to help understand its pros and cons.

**Performance of Ranker AR2-D**: To evaluate the performance of ranker AR2-D on NQ, we first retrieve the top-100 documents for each query in the test set with the help of dual-encoder AR2-G model, and then re-rank them with the scores produced by the AR2-D model. The results are shown in Table 2. "-" represents without ranker. AR2-$D^0$ refers to the warm-up ranker model in AR2. The results show that the ranker obtains better performance compared with only using retriever. It suggests that we could use a two-stage ranking strategy to further boost the retrieval performance. Comparing the results of AR2-D and AR2-$D^0$, we further find that the ranker AR2-D gets a significant gain with adversarial training.

**Impact of Negative Sample Size:** In the training of AR2, the number of negative documents $n$ would affect both the model performance and training time. In Table 3, we show the performance and the training latency per batch with different negative sample size $n$. In this setting, we evaluate AR2 without the regularization term. We observe the improvement over R@1 and R@5 by increasing $n$ from 1 to 7, and marginal improvement when keep increasing $n$ from 7 to 15. The latency of training per batch is almost linear increased by improving $n$.

**Comparison with IRGAN**: The original IRGAN (Wang et al., 2017) doesn't work for dense text retrieval tasks as it does not contain the dual-encoder retrieval model for fast document indexing and search. However, it provides an conventional GAN framework for training the generative and discriminative models jointly for IR tasks. To compare the proposed AR2 with IRGAN, we replaced the generative and discriminative models in IRGAN with the retriever and ranker models in AR2,

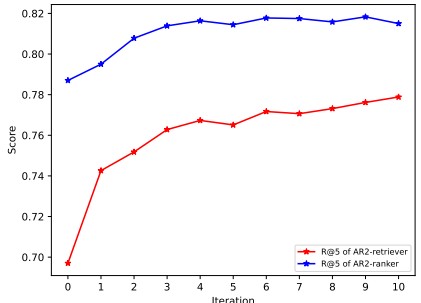 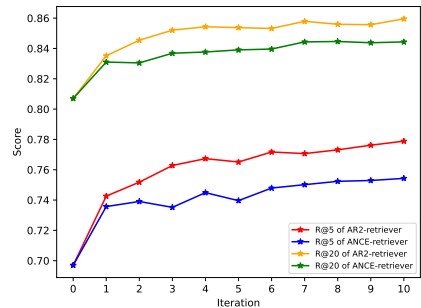

Figure 3: NQ R@5 on the number of iteration for both the AR2-retriever and the AR2-ranker.

Figure 4: The comparison of ANCE and AR2 on NQ test set.

Table 6: The results of the second-stage ranking on Natural Questions test set. Note that we copy the numbers of the first block from the RIDER paper (Mao et al., 2021b).

| Retriever | Ranker | R@1 | R@5 | R@10 | R@20 | R@50 | R@100 |
|---|---|---|---|---|---|---|---|
| GAR$^+$ (Mao et al., 2021a) | - | 46.8 | 70.7 | 77.0 | 81.5 | - | 88.9 |
| GAR$^+$ (Mao et al., 2021a) | BERT | 51.4 | 67.6 | 75.7 | 82.4 | - | 88.9 |
| GAR$^+$ (Mao et al., 2021a) | BART | 55.2 | 73.5 | 78.5 | 82.2 | - | 88.9 |
| GAR$^+$ (Mao et al., 2021a) | RIDER | 53.5 | 75.2 | 80.0 | 83.2 | - | 88.9 |
| AR2-**G** | - | 58.7 | 77.9 | 82.5 | 86.0 | 88.5 | 90.1 |
| AR2-**G** | AR2-**D** | 65.6 | 81.5 | 84.9 | 87.2 | 89.5 | 90.1 |

respectively. Therefore, with the configuration of the same model architectures for generator (retriever) and discriminator (ranker), The performance of the retriever is shown in Table 4. We see that AR2 outperforms IRGAN significantly.

**Effect of Regularization**: To study the effectiveness of regularization, we conducted ablation studies by removing the regularization term in the training of retrieval model. In Table 5, "*R*" refers to the regularization item, it shows that the regularization approach helps to improve the R@1 and R@5 evaluation metrics. In additional, we compute the average entropy of distribution $p_\theta(\cdot|q, d, \mathbb{D}_q)$ on the NQ test set, where $\mathbb{D}_q$ is the retrieved top-15 documents. The average entropy measures the sharpness of distribution $p_\theta(\cdot|q, d, \mathbb{D}_q)$. In experiments, the average entropies for with *R* and w/o *R* in AR2-**G** are 2.10 and 1.70 respectively. This indicates that the regularization term could help smooth the prediction of probabilities in the retriever.

**Visualization of the Training Procedure:** We visualize the changes of R@5 during the AR2-**G** training. The result is shown in Figure 3. We see that as adversarial iteration increases, the R@5 of both AR2-retriever and AR2-ranker also gradually increases. AR2-retriever has the most significant improvement (about 4.5%) after the first iteration. While the training advances closer to the convergence, the improvement of R@5 also gradually slows down. In the end, AR2-retriever is improved by approximately 8% and AR2-ranker is improved by approximately 3%.

**Adversarial Training versus Iterative Hard-Negative Sampling**: To give a fair comparison of AR2 and ANCE (Xiong et al., 2021), we retrain the ANCE model by initializing it with the same warm-up AR2-**G$^0$** which leverages the advantage of the continuous pre-training technique. In experiments, ANCE trains the retriever with an iterative hard-negative sampling approach instead of adversarial training in AR2. In Figure 4, we observe that AR2 steadily outperforms ANCE during training in terms of R@5 and R@10 evaluation metrics with the same model-initialization. It shows that AR2 is a superior training framework compared with ANCE.

**Performance of the Pipeline:** We evaluate the performance of the retrieve-then-rank pipeline on NQ dataset. The results are shown in Table 6. GAR$^+$ is a sparse retriever which ensembles GAR (Mao et al., 2021a) and DPR (Karpukhin et al., 2020). BERT (Nogueira & Cho, 2019), BART (Nogueira et al., 2020), and RIDER (Mao et al., 2021b) are three ranking methods. BERT ranker is a cross-encoder, which makes a binary relevance decision for each query-passage pair.

BART ranker generates relevance labels as target tokens in a seq2seq manner. RIDER re-ranks the retrieved passages based on the lexical overlap with the top predicted answers from the reader. The results show that AR2 pipeline significantly outperforms existing public pipelines.

## 5 RELATED WORK

**Text Retrieval:** Text retrieval aims to find related documents from a large corpus given a query. Retrieval-then-rank is the widely used pipeline (Huang et al., 2020; Zou et al., 2021).

For the first stage retrieval, early researchers used sparse vector space models, e.g., BM25 (Yang et al., 2017). Recently, some works improve the traditional sparse retriever with neural network, e.g., Dai & Callan (2019) use BERT to dynamically generate term weights, doc2Query (Nogueira et al., 2019b), docTTTTTQuery (Nogueira et al., 2019a), and GAR (Mao et al., 2021a) use text generation to expand queries or documents to make better use of BM25.

Recently, dense retrieval methods have become a new paradigm for the first stage of retrieval. Various methods have been proposed to enhance dense retrieval, e.g., DPR (Karpukhin et al., 2020) and ME-BERT (Luan et al., 2021) use in-batch negatives and construct hard negatives by BM25; ANCE (Xiong et al., 2021), RocketQA (Qu et al., 2021), and ADORE (Zhan et al., 2021) improve the hard negative sampling by iterative replacement, denoising, and dynamic sampling, respectively; PAIR (Ren et al., 2021) leverages passage-centric similarity relation into training object; FID-KD (Izacard & Grave, 2020) and RDR (Yang & Seo, 2020) distill knowledge from reader to retriever; Guu et al. (2020) and Sachan et al. (2021b) enhance retriever by jointly training with downstream tasks. Some researchers focus on the pre-training of dense retrieval, such as ICT (Lee et al., 2019), Condenser (Gao & Callan, 2021b) and Cocondenser (Gao & Callan, 2021a).

For the second stage ranking, previous works typically use cross-encoder based methods. The cross-encoder models which capture the token-level interactions between the query and the document (Guo et al., 2016; Xiong et al., 2017), have shown to be more effective. Various methods are proposed to enhance ranker, e.g., Nogueira & Cho (2019) use BERT to make a binary relevance decision for each query-passage pair; Nogueira et al. (2020) adopt BART to generate relevance labels as target tokens in a seq2seq manner; Khattab & Zaharia (2020) and Gao et al. (2020) adopt the lightweight interaction based on the representations of dense retrievers to reduce computation. However, negative samples are statically sampled in these works. In AR2, negative samples for training the ranker will be dynamically adjusted with the progressive retriever.

**Generative Adversarial Nets:** Generative Adversarial Nets (Goodfellow et al., 2014) have been widely studied in the generation field, i.e., image generation (Brock et al., 2018) and text generation (Yu et al., 2017). With a minimax game, GAN aims to train a generative model to fit the real data distribution under the guidance of a discriminative model. Few works study GAN to text retrieval. A related work is IRGAN (Wang et al., 2017). It proposes a minimax retrieval framework that aims to unify the generative and discriminative retrieval models.

## 6 CONCLUSION

In this paper, we introduce AR2, an adversarial retriever-ranker framework to jointly train the end-to-end retrieve-then-rank pipeline. In AR2, the retriever retrieves hard negatives to cheat the ranker, and the ranker learns to rank the collection of positives and hard negatives while providing progressive rewards to the retriever. AR2 can iteratively improve the performance of both the retriever and the ranker because (1) the retriever is guided by the progressive ranker; (2) the ranker learns better through the harder negatives sampled by the progressive retriever. AR2 achieves new state-of-the-art performance on all three competitive benchmarks.

**Acknowledgement**

This work is supported by the National Natural Science Fund for Distinguished Young Scholar (Grant No. 61625204), and partially supported by the Key Program of National Science Foundation of China (Grant No. 61836006).

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

## A APPENDIX

### A.1 PROOF

**Proof of Eqn. 9:** Suppose $d_i^- \in \mathbb{D}_q^-$ is sampled by $p_\theta(\cdot|q; \mathbb{D}_q^-)$, thus

$$
\begin{aligned}
J^\theta &= \mathbf{E}_{\mathbb{D}_q^- \sim G_\theta(q, \cdot)} \left[ \log p_\phi(d|q; \{d\} \cup \mathbb{D}_q^-) \right] \\
&\leq \mathbf{E}_{\mathbb{D}_q^- \sim G_\theta(q, \cdot)} \left( \mathbf{E}_{d_i^- \sim p_\phi(\cdot|q; \mathbb{D}_q^-)} \left[ \log p_\phi(d|q; \{d, d_i^-\}) \right] \right)
\end{aligned}
\tag{13}
$$

where $\mathbb{D}_q^-$ indicates the set of negative documents sampled by $G_\theta(q, \cdot)$. In practice, we approximate $\mathbb{D}_q^-$ by sampling $n$ documents from the top-$K$ retrieved negative set. Therefore, we could further obtain the following approximately equation in implementation.

$$
\approx \mathbf{E}_{d_i^- \sim p_\theta(\cdot|q; \mathbb{D}_q^-)} \left[ \log p_\phi(d|q; \{d, d_i^-\}) \right] = \hat{J}^\theta
\tag{14}
$$

**Proof of Eqn. 10:**

$$
\begin{aligned}
\nabla_\theta \hat{J}^\theta &= \nabla_\theta \mathbf{E}_{d_i^- \sim p_\theta(\cdot|q; \mathbb{D}_q^-)} \left[ \log p_\phi(d|q; \{d, d_i^-\}) \right] \\
&= \sum_i \nabla_\theta p_\theta(d_i^-|q; \mathbb{D}_q^-) \left[ \log p_\phi(d|q; \{d, d_i^-\}) \right] \\
&= \sum_i p_\theta(d_i^-|q; \mathbb{D}_q^-) \nabla_\theta \log p_\theta(d_i^-|q; \mathbb{D}_q^-) \left[ \log p_\phi(d|q; \{d, d_i^-\}) \right] \\
&= \mathbf{E}_{d_i^- \sim p_\theta(\cdot|q; \mathbb{D}_q^-)} \nabla_\theta \log p_\theta(d_i^-|q; \mathbb{D}_q^-) \left[ \log p_\phi(d|q; \{d, d_i^-\}) \right]
\end{aligned}
\tag{15}
$$

### A.2 EFFICIENCY REPORT

We list the time cost of training and inference in Table 7. The evaluation is made with 8 NVIDIA A100 GPUs. The max step of ANCE training is from the ANCE's open-source website [2].We estimate the overall training time without taking account of the time of continuous pre-training step and warming-up step.

Table 7: Comparison of Efficiency

|  | DPR | ANCE | AR2(n=15) | AR2(n=1) |
|---|---|---|---|---|
| **Training** | | | | |
| Batch Size | 128 | 128 | 64 | 64 |
| Max Step | 20k | 136k | 20k | 20k |
| BP for Retriever | 1.8h | 11h | 2.3h | 1h |
| BP for Ranker | - | - | 0.75h | 0.35h |
| Iteration Number | 0 | 10 | 10 | 10 |
| Index Refresh | 0.5 | 0.5h | 0.5h | 0.5h |
| Overall | 1.85h | 16h | 9.1h | 6.4h |
| **Inference** | | | | |
| Encoding of Corpus | 20min | 20min | 20min | 20min |
| Query Encoding | 40ns | 40ns | 40ns | 40ns |
| ANN Index Build | 2min | 2min | 2min | 2min |
| ANN Retrieval(Top-100) | 2ms | 2ms | 2ms | 2ms |

---

[2]https://github.com/microsoft/ANCE

## A.3 HYPERPARAMETERS

Table 8: Hyperparameters for AR2 training.

|  | Parameter | NQ | TriviaQA | MS-MARCO |
|---|---|---|---|---|
| Default | Max query length | 32 | 32 | 32 |
|  | Max passage length | 128 | 128 | 128 |
| AR2-$G^0$ | Learning rate | 1e-5 | 1e-5 | 1e-4 |
|  | Negative size | 255 | 255 | 127 |
|  | Batch size | 128 | 128 | 64 |
|  | Temperature $\tau$ | 1 | 1 | 1 |
|  | Optimizer | AdamW | AdamW | AdamW |
|  | Scheduler | Linear | Linear | Linear |
|  | Warmup proportion | 0.1 | 0.1 | 0.1 |
|  | Training epoch | 40 | 40 | 3 |
| AR2-$D^0$ | Learning rate | 1e-5 | 1e-5 | 1e-5 |
|  | Negative size | 15 | 15 | 15 |
|  | Batch size | 64 | 64 | 256 |
|  | Temperature $\tau$ | 1 | 1 | 1 |
|  | Optimizer | AdamW | AdamW | AdamW |
|  | Scheduler | Linear | Linear | Linear |
|  | Warmup proportion | 0.1 | 0.1 | 0.1 |
|  | Training step per iteration | 1500 | 1500 | 1500 |
|  | Max step | 2000 | 2000 | 4000 |
| AR2-$G$ | Learning rate | 1e-5 | 1e-5 | 5e-6 |
|  | Negative size | 15 | 15 | 15 |
|  | Batch size | 64 | 64 | 64 |
|  | Temperature $\tau$ | 1 | 1 | 1 |
|  | Optimizer | AdamW | AdamW | AdamW |
|  | Scheduler | Linear | Linear | Linear |
|  | Warmup proportion | 0.1 | 0.1 | 0.1 |
|  | Training step per iteration | 1500 | 1500 | 1500 |
|  | Max step | 15000 | 15000 | 15000 |
| AR2-$D$ | Negative size | 15 | 15 | 15 |
|  | Learning rate | 1e-6 | 1e-6 | 5e-7 |
|  | Batch size | 64 | 64 | 64 |
|  | Temperature $\tau$ | 1 | 1 | 1 |
|  | Optimizer | AdamW | AdamW | AdamW |
|  | Scheduler | Linear | Linear | Linear |
|  | Warmup proportion | 0.1 | 0.1 | 0.1 |
|  | Training step per iteration | 500 | 500 | 500 |
|  | Max step | 5000 | 5000 | 5000 |

### A.4 MODEL CONFIGURATION AND EXPERIMENT SETTINGS

We list the detailed configuration of AR2 and baseline models in Table 9.

Table 9: Model configuration and experiment settings.

| Model | Initial Model | Parameters | Further Pretrain | Additional Data |
|---|---|---|---|---|
| DPR (Karpukhin et al., 2020) | BERT-Base | 110M | - | - |
| ANCE (Xiong et al., 2021) | BERT/RoBERTa-Base | 110M/125M | - | - |
| RocketQA (Qu et al., 2021) | ERNIE-2.0-Base
ERNIE-2.0-Large | 110M
330M | -
- | 1.7 M |
| PAIR (Ren et al., 2021) | ERNIE-2.0-Base
ERNIE-2.0-Large | 110M
330M | -
- | 1.7 M |
| Individual Top-k (Sachan et al., 2021a) | ERNIE-2.0-Base
T5-Large | 110M
739M | Yes
- | - |
| coCondenser (Gao & Callan, 2021a) | BERT-Base | 110M | Yes | - |
| Our (AR2-G) (Retriever)
Our (AR2-D) (Ranker) | ERNIE-2.0-Base
ERNIE-2.0-Large | 110M
330M | Yes
- | -
- |

### A.5 ABLATION STUDY ON DIFFERENT INITIAL MODELS

Table 10 shows the results of our method with different initial models. We see that ERNIE-Base as the initial model achieves a little better performance than BERT-Base. And AR2-G using BERT-Base as the initial model still achieves better performance than other methods under the same initial model. Meanwhile, ICT pre-training improves the performance of AR2-G.

Table 10: Performance of AR2-**G** on NQ test set with different initial model

| | Initial Model | R@1 | R@5 | R@20 | R@100 |
|---|---|---|---|---|---|
| DPR (Karpukhin et al., 2020) | BERT-Base | - | - | 78.4 | 85.3 |
| ANCE (Xiong et al., 2021) | BERT-Base | - | - | 81.9 | 87.5 |
| RocketQA (Qu et al., 2021) | ERNIE-Base | - | 74.0 | 82.7 | 88.5 |
| PAIR (Ren et al., 2021) | ERNIE-Base | - | 74.9 | 83.5 | 89.1 |
| AR2-G | BERT-Base | 56.7 | 76.1 | 85.0 | 89.3 |
| AR2-G | ERNIE-Base | 57.2 | 76.6 | 85.3 | 89.8 |
| AR2-G | ERNIE-Base w/ ICT | 58.7 | 77.9 | 86.0 | 90.1 |

## A.6 COMPARISON WITH SEVERAL EXISTING APPROACHES

Table 11 shows the comparison of AR2 and several existing retrieval approaches. "Extra Label" refers to whether the answer label is used. AR2 jointly optimizes both the retriever and the ranker according to a principle adversarial objective, which is the key difference with previous works.

Table 11: Comparison with existing approaches

| Model | Extra Label | Retriever-Ranker/ Retriever-Reader | Adversarial Objective | Update Hard Negatives |
|---|---|---|---|---|
| FID-KD (Izacard & Grave, 2020) | Yes | Yes | No | No |
| RDR (Yang & Seo, 2020) | Yes | Yes | No | No |
| RocketQA (Qu et al., 2021) | No | Yes | No | Yes |
| ANCE (Xiong et al., 2021) | No | No | No | Yes |
| RIDER (Mao et al., 2021b) | Yes | Yes | No | No |
| AR2 | No | Yes | Yes | Yes |

## A.7 PERFORMANCE OF THE PIPELINE

Table 12 shows the performance of the retrieve-then-rank pipeline on Trivia QA and MS-MARCO. From the results of Table 6 and Table 12, we find that the ranker AR2-D improves the performance on all three benchmarks including NQ, Trivia QA, and MS-MARCO. Meanwhile, the pipeline based on AR2 achieves state-of-the-art performances on all benchmarks.

Table 12: The results of the second-stage ranking on Trivia QA and MS-MARCO.

| Retriever | Ranker | Trivia QA | | | | MS-MARCO |
|---|---|---|---|---|---|---|
| | | R@1 | R@5 | R@10 | R@20 | MRR@10 |
| RepBERT (Zhan et al., 2020) | RepBERT (Zhan et al., 2020) | - | - | - | - | 37.7 |
| ME-HYBIRD (Luan et al., 2021) | ME-HYBIRD (Luan et al., 2021) | - | - | - | - | 39.4 |
| ME-BERT (Luan et al., 2021) | ME-BERT (Luan et al., 2021) | - | - | - | - | 39.5 |
| BM25 (Yang et al., 2017) | TFR-BERT (Han et al., 2020) | - | - | - | - | 40.5 |
| GAR$^+$ (Mao et al., 2021a) | RIDER (Mao et al., 2021b) | 71.9 | 77.5 | 79.8 | 81.8 | - |
| AR2-G | - | 64.2 | 78.2 | 81.8 | 84.4 | 39.5 |
| AR2-G | AR2-D | 73.0 | 82.1 | 84.1 | 85.8 | 43.2 |

## A.8 PERFORMANCE OF THE LARGE-SIZE MODEL

Table 13 shows the results of AR2-G (Retriever) initialized with ERNIE-2.0-Large (without continuous pre-training (ICT)). All baselines are initialized by large-size model, and DPR-PAQ (Oğuz et al., 2021) utilizes a large external corpus (65m question-answer pairs) to continue pre-training the model; Individual Top-K (Sachan et al., 2021a) utilizes T5-Large model (739M parameters vs 330M parameters ERNIE-2.0-Large) as reader to guide the retriever. Compared with these baseline methods, AR2-G achieves a significant performance improvement, which further demonstrates the effectiveness of AR2-G (Retriever).

Table 13: The performance of large-size models on Natural Questions test set,

| | Size | R@1 | R@5 | R@20 | R@100 |
|---|---|---|---|---|---|
| DPR-PAQ$_{BERT}$ (Oğuz et al., 2021) | Large | - | 75.3 | 84.4 | 88.9 |
| DPR-PAQ$_{RoBERTa}$ (Oğuz et al., 2021) | Large | - | 76.9 | 84.7 | 89.2 |
| Individual Top-K (Sachan et al., 2021a) | Large | 57.5 | 76.2 | 84.8 | 89.8 |
| AR2-G | Base | 58.7 | 77.9 | 86.0 | 90.1 |
| AR2-G | Large | 61.1 | 78.8 | 86.5 | 90.4 |

