# OpenReview forum: "Adversarial Retriever-Ranker for Dense Text Retrieval"
_ICLR.cc/2022/Conference — ICLR 2022 Poster_

### Official Review · Reviewer_dE9M · 2021-11-01

**Correctness:** 3
**Technical Novelty And Significance:** 3
**Empirical Novelty And Significance:** 2
**Recommendation:** 6
**Confidence:** 4

**Main Review:**

This paper proposes a new method to improve deep retrieval performance. The key idea is to use the ranker to help retriever training. The key novelty of this paper is to formulate the problem as an adversarial training procedure. After some approximations, the framework essentially use the ranker as a reward function and a teacher model to distill from. Experiments show better retrieval performance than existing methods on some popular benchmark datasets.

Strength:
- The reviewer feels the adversarial formulation is interesting and intuitive.
- The retrieval performance is good and seem to achieve state-of-art numbers.

Weakness:
- The major concern of this paper is the comprehensiveness of final ranking performance. Though the retrieval numbers are good, it is not clear why the numbers on ranking tasks are significantly lacking. Specifically, only recall numbers on the NQ dataset is reported. Please report reranking numbers with more metrics (such as MRR) on all datasets. For example, in the RocketQA paper, they explicitly focus on the retrieval but still report final ranking performance combined with different re-rankers. This paper focuses a lot of "retriever-reranker" joint optimization, so the authors should either 1) Make the contribution clear that this paper mainly focus on retriever. But that will limit the contribution of this paper. Also it is not clear why the "retriever-reranker" joint optimization cannot achieve state-of-art ranking performance. If that is the case, the impact of the retriever is not clear in real-world applications. 2) Or stick to the joint optimization framework, in which case user-facing ranking performance is arguably more important than retrieval performance anyway.
- Novelty may not be very significant. Using ranker to help retriever is not a new idea. This paper's main contribution is a more principled loss. The significance of contribution is subjective and the reviewer is generally ok with it. But still feel it needs support from more comprehensive evaluations to meet the bar of top-tier venues.



**Summary Of The Paper:**

This paper proposes to use adversarial loss and distillation loss to improve dense retriever performance from the help of the ranker. Experiments show better retrieval performance than existing methods.

**Summary Of The Review:**

The paper proposes a new loss function to leverage ranker to help retriever for text retrieval tasks. Though the idea is interesting, the evaluations now lack important components so the significance of this work is not very convincing. The authors are suggested to provide more details in the rebuttal.

---

> ### Author Response · Authors · 2021-11-14
> **Response to Reviewer dE9M**
>
> Thanks for your valuable comments.
> > *Q1*:  Why the numbers on ranking tasks are significantly lacking. Please report reranking numbers with more metrics (such as MRR) on all datasets.
>
> In Table 7 and Table 3, we have reported the performance of AR2-D (ranker) on NQ dataset to compare it with existing retrieval-then-rerank approaches. The detailed analysis is given in Section 4.4 (in the subsection “Performance of Ranker AR2-D”  and subsection "Performance of the Pipeline").  Following your valuable suggestion about a more comprehensive study, we performed two additional studies using MS-MARCO and TriviaQA on the end-to-end ranking tasks. Following table shows the performance of our method. We added a more detailed comparison of baseline methods and our model to the revision (in Appendix.A7) as your suggestion.
>
>  Performance of AR2-D (Ranker) on NQ, MS-MARCO, and TriviaQA.
>
> |                   | MS-MARCO MRR@10 | TriviaQA R@5 | TriviaQA R@20 | NQ R@5 | NQ R@20 |
> | ----------------- | --------------- | ------------ | ------------- | ------ | ------- |
> | AR2-G (Retriever) | 39.5            | 78.2         | 84.4          | 77.9   | 86.0    |
> | AR2-D (Ranker)    | 43.2            | 82.1         | 85.5          | 81.5   | 87.2    |
>
> Please let me know if anything is unclear or if you have any additional concerns.

---

### Official Review · Reviewer_idsf · 2021-11-02

**Correctness:** 3
**Technical Novelty And Significance:** 3
**Empirical Novelty And Significance:** 2
**Recommendation:** 6
**Confidence:** 4

**Main Review:**

Strengths:

1. The paper proposed a (novel) approach for multi-stage training of a retriever and ranker model in an adversarial manner. To the best of my knowledge, previous approaches using a retriever and a ranker were trained in a pipelined manner but not using the minimax adversarial objective. The adversarial training of the model makes the proposed model different from the previous work.

2. The model outperforms a number of recent baselines on three benchmark text retrieval tasks. As these baselines are highly competitive, I believe that the improvements are significant.

Weaknesses:

1. The paper does not discuss the reasons for selecting the large configuration for the ranker and base configuration for the retriever. In my opinion, the large configuration of the ranker contributes a lot to the performance gains. Most of the previous approaches in their Table 2 consist of just the base configuration of models.
The large configurations has an increased number of layers due to which the representational capability of the ranker increases and thus  comparisons with the base models is not fair.
It would be nice if in the rebuttal the authors can disentangle the effect of configuration choice. For example, having two results tables or sections with just the base configuration and large configuration and discuss their results. For large configurations, the authors should then compare with the respective results of the baseline models.

2. The authors should report the total number of parameters in the model the in the baseline models. This will help in better understanding of the pros and cons of the different approaches. I believe that this should be easy to do.

3. There have been models proposed in previous work that consist of the retriever and ranker components. While it is great that the authors have cited most of the previous work, they should explain in detail the difference between their work and such previous work. For example, there can be a table which contains salient aspects of different models.

Questions for the authors: Can you include the results of these new experiments in the author's response?

1. Results when just initializing the retriever with either ERNIE or ICT and not applying DPR on them? This will shed light on how much the retriever is robust to initialization parameters. What's the rationale for initializing the model with ERNIE rather than with BERT?

2. How is the D_q^(-) computed for training the retriever? Is it done in the same manner as for the reranker? What happens if we take all the documents in top-100 and don't sample from them?

3. Would it be possible to perform end-to-end training in a single stage instead of the multi-stage step-wise training and then compare their results?

4. I feel that with the current description of the model, it is hard to interpret what the retriever has learned. The weight update step looks similar to the REINFORCE algorithm. Is the retriever training sensitive to hyperparameter choices ?

**Summary Of The Paper:**

This paper proposes a minimax multi-stage iterative method for document retrieval. The model consists of two components: dense retriever and ranker. The dense retriever is modeled using a dual-encoder and the ranker is modeled with a cross-encoder, both of which are fairly standard. In their proposed approach, in the first step, the ranker is trained using a contrastive loss defined over the top-K documents from the pre-trained retriever. In the second step, the retriever is trained using an objective similar to REINFORCE to maximize the likelihood of the negative documents. Such stepwise training forces the ranker to improve its predictions because the adversarial training of the retriever leads to more hard-negatives in the top-K documents. The authors perform experiments on three widely used datasets for ranking tasks and compare with a number of recent baselines. Their results show good gains in retrieval recall over the prior models, which can be significant. The authors also include ablation studies to better understand their proposed approach.

**Summary Of The Review:**

Summarizing from above:

The proposed approach has novel aspects that is also demonstrated by its state-of-the-art results on benchmark IR tasks. On the other hand, I expect the authors to conduct more comprehensive experiments to facilitate fair comparisons with baselines, which I have also mentioned under weakness above. The authors should uniformly select the model configurations and compare with the respective baselines from the literature. Other than this, I have several questions regarding the model initialization and training process.
Answering them will also help the readers develop a more coherent understanding of the paper.

Depending on the author's response, I am open to reconsidering my scores.

[Update]
I want to thank the authors for doing additional experiments and answering the questions raised in the reviews. I am happy with the inclusion of the new results and so I am increasing my scores to 6.

---

> ### Author Response · Authors · 2021-11-14
> **Response to Reviewer idsf (Part 2)**
>
> > *Q6*: Would it be possible to perform end-to-end training in a single stage instead of the multi-stage step-wise training and then compare their results?
>
> We are not sure whether we understand your question about "single stage end-to-end training" correctly here. Mathematically, AR2 belongs to the end-to-end training paradigm. However in the practical implementation, since we need to update the retriever-index after each training stage of AR2-G, jointly optimizing the retriever-ranker end-to-end during training is challenging.
>
> > *Q7*: I feel that with the current description of the model, it is hard to interpret what the retriever has learned. The weight update step looks similar to the REINFORCE algorithm. Is the retriever training sensitive to hyperparameter choices ?
>
> 1.	In Section 4.4 (page 8, last paragraph), we show the performance curve of both the retriever and the ranker during the multi-stage training. It shows the performance of both the ranker and the retriever is gradually improved during training.
> 2.	The update formulation of retriever in Eqn. (10) is closely related to REINFORCE algorithm, where the ranker assigns “rewards” to retrieved examples.
> 3.	The training of retriever is not sensitive to learning-rate/batch size in our experiments.
>
> Please let me know if anything is unclear or if you have any additional concerns.

---

> ### Author Response · Authors · 2021-11-14
> **Response to Reviewer idsf (Part 1)**
>
> Thanks for your valuable comments.
> > *Q1*:
> > a). Most of the previous approaches in their Table 2 consist of just the base configuration of models. The large configurations has an increased number of layers due to which the representational capability of the ranker increases and thus comparisons with the base models is not fair.
> >  b). The reasons for selecting the large configuration for the ranker and base configuration for the retriever.
> >  c). The effect of configuration, For example, having two results tables or sections with just the base configuration and large configuration and discuss their results. For large configurations, the authors should then compare with the respective results of the baseline models.
>
> a). All the models and results in Table 2 are based on base-size configuration, including our AR2-G. Since they are similar size, we believe these comprehensive comparisons are fair. The performance of the ranker with large-size configuration is reported In Table 7 Section 4.4.
> b). The reason we chose different models for the retriever and the ranker is to follow previous works for a fair comparison. We provide the detailed configuration to illustrate this in the General Response (#1).
> c). Thanks for the suggestion. Note that most previous works, including RocketQA (Qu et al, 2021), PAIR(Ren et al, 2021), and ANCE (Xiong et al, 2021),  only provide their base-size numbers in their retriever-only experiments, so thus it is hard for us to compare with them in the large-size models. We will try to provide an additional experiment on large-size in the final revision.
>
> > *Q2*: The authors should report the total number of parameters in the model and in the baseline models.
>
> We provide an answer in the General Response (#1) to your question. We make this clear in the revision.
>
> > *Q3*: The difference between AR2 and previous works that consist of the retriever and ranker components
>
> Please see it in the General Response (#3). In summary, AR2 jointly optimizes the two modules according to an adversarial objective. During the adversarial procedure, the retriever learns to retrieve negative documents to cheat the ranker, while the ranker learns to rank a collection of candidates including both the ground-truth and the retrieved ones, as well as providing progressive direct feedback to the retriever.
>
> > *Q4*: Results when just initializing the retriever with either ERNIE or ICT and not applying DPR on them? This will shed light on how much the retriever is robust to initialization parameters.  What's the rationale for initializing the model with ERNIE rather than with BERT?
>
> 1. Following table shows the performance without warmup. Without DPR warmup would affect the performance of AR2, but it could still achieve reasonable good performance compared with other baseline approaches. We think the comparison here is still fair since all the baseline models have the DPR warmup step.
> |                                             | R@1  | R@5  | R@20 | R@100 |
> | ------------------------------------------- | ---- | ---- | ---- | ----- |
> | AR2-G                                       | 58.7 | 77.9 | 86.0 | 90.1  |
> | AR2-G w/o warm up  | 55.4 | 76.0 | 84.8 | 89.5  |
> 2. Please see the updated experimental results in General Response (#2). It shows that initializing with ERNIE is slightly better than BERT.
>
> > *Q5*: How is the $D_q^{-}$ computed for training the retriever? Is it done in the same manner as for the reranker? What happens if we take all the documents in top-100 and don't sample from them?
>
> 1.  Yes, $D_q^{-}$ is computed in the same manner for retriever and ranker training.
> 2.  In Table 4 and Section 4.4 (page 7), we give the analysis of choosing different numbers of negative samples in training.  More negative samples would significantly increase the computation cost. We have not tried to use all 100 samples as negatives, If you think it is necessary, we will try it.

---

> > ### Author Response · Authors · 2021-11-19
> > **Further explanation of Q5**
> >
> > > *Q*: What happens if we take all the documents in Top-100 as negatives and don’t sample from them.
> >
> > Thanks for your suggestion. We conduct this experiment, the following table shows the results, we see that the performance under the two settings is comparable, but the sampling strategy is more efficient for training (about 5 times in our experiment).
> >
> > |                             | R@1  | R@5  | R@20 | R@100 |
> > | --------------------------- | :----: | :----: | :----: | :-----: |
> > | AR2-G (Sample 15 negatives) | 58.7 | 77.9 | 86.0 | 90.1  |
> > | AR2-G (Take all Top-100)    | 58.0 | 78.4 | 85.9 | 90.4  |

---

### Official Review · Reviewer_ySzP · 2021-11-02

**Correctness:** 4
**Technical Novelty And Significance:** 3
**Empirical Novelty And Significance:** 3
**Recommendation:** 8
**Confidence:** 5

**Main Review:**

Compared to the most related work (IRGAN), the main differences are (1) the use of a contrastive loss rather than a cross-entropic one, (2) a different pre-training, inspired by most recent works, (3) regularization through distillation (KL between retriever and ranker), (4) a different sampling procedure. These differences taken together are substantial, and bring a better performance. I think the paper does a good job at showing that all the propositions are complementary.

The section 3 describes the training procedure. I found the notation $p(d|q,d..)$ a bit misleading because $d$ appears on both side, but besides this, the section reads easily.

Experiments (section 4) are conducted on the MS Marco, Trivia QA, and Natural Questions, showing that the retriever performs better that state-of-the-art baselines. The analysis section is quite interesting, giving insight on various aspects of the model/framework (including the regularization procedure, the improvement brought by the adversarial optimization, and an interesting comparison with other related models e.g. IRGAN and ANCE). It succeeds in showing that all the presented techniques, including the main proposition (adversarial training procedure) do indeed allows the model to perform better than other propositions.

The experiments conclude with results for the full pipeline in the case of Natural Questions. It would have been interesting to look the results for the other collections (MS Marco and Trivia QA) - especially for MS Marco, since most models are evaluated on the full pipeline.

The appendix A.3 gives all the hyperparameter settings which is important for reproducibility.

Other remarks

- the English should be revised in many places (improper use of word “it”, “on the other hand” 3rd paragraph 2nd page)
- Please be clear when you compare your models with 1st stage rankers (e.g. table 2) - since looking at the results from the papers
- The notation $p(d|q,d,\mathbb D^-_q)$ is a bit weird: use a different notation for the first $d$
- Appendix A.1: the first inequality of eq. (13) is not obvious unless the probability is computed through a softmax. The next « approximately equal » is not really proven (is it really approximately equal since documents are independent ?)

**Summary Of The Paper:**

This paper proposes to learn a retriever (to find relevant documents in a dataset / dual encoder) and a ranker (to re-rank retrieved documents / cross-encoder) using an adversarial framework (the retriever trying to fool the ranker). This training allows to reach state-of-the-art performance for both the retriever and the ranker. The paper leverages all the recent techniques of neural retrieval models (pre-training, teacher training) and paper mostly focus on the training of the 1st stage (retriever), but also presents result on the full pipeline too (the 2nd stage is more standard, so it is normal the authors did not focus on this part).

Overall, this is a good paper that is inspired by many recent techniques on neural retrievers, bringing back the IRGAN adversarial training but with more success. The experiments are well designed and the analysis interesting.

**Summary Of The Review:**

The paper is an interesting proposition to train retriever-reranker couple for neural information retrieval within a single adversarial framework. The paper demonstrates state-of-the-art results and provide an interesting analysis of the model. There are some minor problems with the paper, but overall I recommend acception.

---

> ### Author Response · Authors · 2021-11-14
> **Response to Reviewer ySzP**
>
> Thanks for your valuable comments.
>
> > *Q1*:  The notation $p(d | q, d, \mathbb{D^-})$ is a bit weird: use a different notation for the first $d$.
>
> Thanks for pointing this out. We have followed your suggestion to improve the notations in the revised paper.
>
> > *Q2*: Appendix A.1: the first inequality of eq. (13) is not obvious unless the probability is computed through a softmax. The next « approximately equal » is not really proven (is it really approximately equal since documents are independent ?)
>
> 1. Yes, the probability $p_{\phi}$ is calculated through a softmax.
> 2. “approximately equal in Eqn. 13”. We think it is rigorous (really) approximately equal given the approximation of $\mathbb{D^-_q}$. We have updated Appendix A.1 to make the proof more readable in the revised paper.
>
> Please let me know if anything is unclear or if you have any additional concerns.

---

### Official Review · Reviewer_hfGT · 2021-11-03

**Correctness:** 3
**Technical Novelty And Significance:** 2
**Empirical Novelty And Significance:** Not applicable
**Recommendation:** 5
**Confidence:** 4

**Main Review:**

Strengths
* The description of the model is generally well-written and is easy to follow.
* Experiments are comprehensive --- evaluated on three well-studied datasets, compared with a range of recent prior work, and analysis showing impact of core components like iterative training, effect of regularization and the effect of number of negatives. The empirical improvements over a range of competitive baselines are also very impressive.

Weaknesses:
* My most critical concern is that there are many components in the paper that are very similar or are already introduced in prior published work, and were not properly credited nor compared.
     * Adversarially and iteratively obtaining hard negatives, which is the core idea of the paper, is explored in prior work including Xiong et al, Oguz et al and Qu et al.
        * The paper seems to be aware of some of these works (briefly mentioned in Section 2). However, their differences are not discussed.
        * I can tell one difference is whether the negatives are coming from its own model (dual encoder) or a separate cross-encoder model, but to me it is a small difference and should have been discussed.
        * Empirical comparison is not provided either - the paper compared with numbers taken from the original papers, but they use different base models so the comparison is unfair.
        * Moreover, some descriptions of prior work in Section 2 are incorrect or at least unclear --- for example, Section 2 says prior work takes negatives from the last checkpoint, indicating a one-time update. However, prior work including Xiong et al and Oguz et al clearly describe that they iteratively choose negatives, as the proposed model in this paper does.
    * Distillation from the ranker to the retriever is identical to methods in prior work including Izacard & Grave and Yang & Seo, and are not cited nor mentioned in the paper.
* The base model used in the paper is ERNIE 2.0 with Inverse Cloze Task and coCondenser, while most prior work uses BERT base. This makes it hard to compare the performance with prior work in a fair manner. (Also, is there any justification for using a different base model for different datasets? Are the decisions made based on the result on the validation data?)
* Given that the base model is different from prior work, one important ablation that is missing in the paper is comparison to the method that iteratively chooses negatives from its own dense model, as in Xiong et al.

Minor comments:
* In Implementation details, it is worth mentioning which text corpus is used for retrieval.


References:
- Xiong et al. ICLR 2021. https://arxiv.org/pdf/2007.00808.pdf
- Oguz et al. 2021. https://arxiv.org/pdf/2012.14610.pdf
- Qu et al. NAACL 2021. https://arxiv.org/pdf/2010.08191.pdf
- Izacard & Grave. ICLR 2021. https://openreview.net/forum?id=NTEz-6wysdb



**Summary Of The Paper:**

This paper proposes an adversarial method to jointly train the dual encoder retriever and the cross-attention ranker. The key idea is to obtain harder negatives for the retriever by training a separate ranker with higher capacity and taking passages that the model is confused with. It also proposes a technique like distillation from the ranker to the retriever. Experiments on three datasets demonstrate improvements over a range of baselines.

**Summary Of The Review:**

To summarize, this paper includes a thorough exploration of better training of dual encoder model using harder negatives chosen in an interactive manner by training a separate, cross-attention model. There has been a number of prior work exploring similar approaches (which I pointed out that comparisons/discussions are not sufficiently provided), but this work is unique in using a cross-attention model for choosing negatives instead of choosing from its own dense retrieval.

There are a few critical concerns: (1) discussions of/comparisons to prior work, (2) justification of using a different base model (as it makes the whole comparison with baselines unfair), and (3) empirical comparison to prior work that uses iterative negatives --- this is an important ablation given that this is a core distinction from prior work as well as the main claim in the paper. They may be relatively easy to be added in the rebuttal so I'd love to increase the score based on the author responses.

---

> ### Author Response · Authors · 2021-11-14
> **Response to Reviewer hfGT**
>
> Thanks for your valuable comments.
>
> > *Q1*: Discussions of prior work
> a). Adversarially and iteratively obtaining hard negatives, which is the core idea of the paper, is explored in prior work including Xiong et al, Oguz et al and Qu et al.
> b). Some descriptions of prior work in Section 2 are incorrect or at least unclear --- for example, Section 2 says prior work takes negatives from the last checkpoint, indicating a one-time update.
> c). Distillation from the ranker to the retriever is identical to methods in prior work including Izacard & Grave and Yang & Seo, and are not cited nor mentioned in the paper.
>
> a). Compared with iterative negative sampling approaches (i.e., ANCE), AR2 jointly optimizes both the retriever and the ranker according to a principle minimax adversarial objective, while ANCE only uses contrastive objective. Their objectives are different in essential. Other differences you can refer to the table in the General Response (#3).
> b). The sentence in section 2 “takes negatives from the last checkpoint”  means “iteratively takes negatives based on previous last checkpoint” not one-time update. We have rephrased this sentence and made it more clear in the revision.
> c). We have cited the two works (Izacard & Grave, 2021 and Yang & Seo, 2020) in section 5 (page 9). We list their differences in the General Response (#3), and we will add more discussions about these works.
>
> > *Q2*: Justification of using a different base model. The base model used in the paper is ERNIE 2.0 with Inverse Cloze Task and coCondenser, while most prior work uses BERT-base. This makes it hard to compare the performance with prior work in a fair manner. (Also, is there any justification for using a different base model for different datasets? Are the decisions made based on the result on the validation data?)
>
> Thanks for your suggestion, here we list the model configurations of AR2 and other baseline approaches in the general response (#1). In the general response (#2), we report the performance of AR2-G with BERT-Base initialization. We have added these tables to our revision. The base models are selected based on the result of the valid set.
>
> > *Q3*: Empirical comparison to prior work that uses iterative negatives. Given that the base model is different from prior work, one important ablation that is missing in the paper is comparison to the method that iteratively chooses negatives from its own dense model, as in Xiong et al.
>
> An empirical comparison between iterative negative sampling (ANCE, Xiong et al 2021) and AR2 has been provided in Figure 4 and section 4.4 (page 8, in the subsection "Adversarial Training versus Iterative Hard-Negative Sampling").
>
> Please let me know if you have any unclear things or additional concerns.

---

### Author Response · Authors · 2021-11-14
**Summary of Paper Revisions**

We thank the reviewers for their constructive feedback, insightful comments and helpful suggestions, and their generally positive appraisal of our work.
- Summary of paper revisions:
  1. We modified some expressions to make the paper more clear as suggested by reviewers.
  2. We added several sections in the appendix, including:
       a). a table to introduce the detailed model configuration of AR2 and its baseline models.
       b). new experiments for AR2-G built on top of BERT-Base and ERNIE-2.0-Base to measure the impacts of different initial pre-trained models.
       c). comparison with existing approaches.
       d). the performance of the second-stage ranking on TriviaQA and MS-MARCO.
       e). the performance of large-size models on Natural Questions.
   3. We added the reference (Oguz et al. 2021) in Section 5 according to the suggestion of reviewers and removed the duplicate references.

---

### Author Response · Authors · 2021-11-14
**General Response**

We thank the reviewers for their constructive feedback, insightful comments and helpful suggestions, and their generally positive appraisal of our work.  We provide a general response to questions raised in several reviews. Responses to individual comments are posted as replies to each review.

1. Model configuration for AR2 and its baseline models

| Model            | Initial model                                         | Parameters  | Further Pretrain | Additional Data |
| ---------------- | ----------------------------------------------------- | :-----------: | :----------------: | :---------------: |
| DPR              | BERT-Base                                             | 110M        | -                | -               |
| ANCE             | BERT-Base (NQ, TriviaQA), RoBERTa-Base (MS-MARCO)     | 110M,125M   | -,-              | -               |
| RocketQA         | ERNIE-2.0-Base (Retriever) / ERNIE-2.0-Large (Ranker) | 110M / 330M | -/-              | 1.7 M           |
| PAIR             | ERNIE-2.0-Base (Retriever) / ERNIE-2.0-Large (Ranker) | 110M / 330M  | -/-              | 1.7 M           |
| Individual Top-k | BERT-Base / T5-Large (Reader)                         | 110M / 739M | Yes/-            | -               |
| Co-condenser     | BERT-Base                                             | 110M        | Yes              | -               |
| AR2 (this work)  | ERNIE-2.0-Base (Retriever) / ERNIE-2.0-Large (Ranker) | 110M / 330M  | Yes/-            | -               |

2. Ablation study on different initial models:
    a. Different initialization using ERNIE-2.0-Base vs. BERT-Base on the NQ dataset. We see that ERNIE-Base as the initial model achieves a little better performance than BERT-Base. And AR2-G using BERT-Base as the initial model still achieves better performance than other methods under the same initial model.
    b. Meanwhile, ICT pre-training improves the performance of AR2-G.

|                                                  | R@1  | R@5  | R@20 | R@100 |
| ------------------------------------------------ | ---- | ---- | ---- | ----- |
|DPR (BERT-Base)     | -    | -    | 78.4 | 85.3  |
| ANCE (BERT-Base)     | -    | -    | 81.9 | 87.5  |
| RocketQA  (ERNIE-2.0-Base)  | -    | 74.0 | 82.7 | 88.5  |
|  PAIR      (ERNIE-2.0-Base) | -    | 74.9 | 83.5 | 89.1  |
|AR2-G (ERNIE-2.0-Base)   | 57.2 | 76.6 | 85.3 | 89.8  |
|  AR2-G (BERT-2.0-Base)  | 56.7 | 76.1 | 85.0 | 89.3  |
| AR2-G (ERNIE-Base w/ ICT)                        | 58.7 | 77.9 | 86.0 | 90.1  |

3. Comparison with existing approaches.

The following table shows the comparison of AR2 and several existing retrieval approaches.  “Extra Label” refers to whether the answer label is used.  AR2 jointly optimizes both the retriever and the ranker according to a principle adversarial objective, which is the key difference with previous works.

| Model                        | Extra Label | Retriever-Ranker / Retriever-Reader | Adversarial Objective | Update Hard Negative |
| ---------------------------- | :-----------: | :-----------------------------------: | :---------------------: | :--------------------: |
| FID-KD (Izacard, Grave 2021) | Yes         | Yes                                 | No                    | No                   |
| RDR  (Yang et al, 2020)      | Yes         | Yes                                 | No                    | No                   |
| RocketQA (Qu et al, 2021)    | No          | Yes                                 | No                    | Yes                  |
| ANCE (Xiong et al, 2020)     | No          | No                                  | No                    | Yes                  |
| RIDER (mao et al, 2020)      | Yes         | Yes                                 | No                    | No                   |
| AR2 (this work)              | No          | Yes                                 | Yes                   | Yes                  |

---

> ### Author Response · Authors · 2021-11-19
> **Performance of Large Model**
>
> Dear reviewers:
>
> We conduct an experiment about large-size models on NQ and we have added it to our paper revision.
>
> Following table shows the results of AR2-G (Retriever) initialized with ERNIE-2.0-Large (without continuous pre-training (ICT)). All baselines in the following table are initialized by large-size model, and DPR-PAQ (Oguz et al. 2021) utilizes a large external corpus (65m question-answer pairs) to continue pre-training the model; Individual Top-K (Sachan et al. 2021) utilizes T5-Large model (739M parameters vs 330M parameters ERNIE-2.0-Large) as reader to guide the retriever. Compared with these baseline methods, AR2-G achieves a significant performance improvement, which further demonstrates the effectiveness of AR2-G (Retriever).
>
> |                                       | Size  | R@1  | R@5  | R@20 | R@100 |
> | ------------------------------------- |:-----:|:----:| :----: |:----:|:-----:|
> | DPR-PAQ(BERT) (Oguz et al. 2021)       | Large |  -   | 75.3 | 84.4 | 88.9  |
> | DPR-PAQ(RoBERTa)  (Oguz et al. 2021)  | Large |  -   | 76.9 | 84.7 | 89.2  |
> | Individual Top-K (Sachan et al. 2021) | Large | 57.5 | 76.2 | 84.8 | 89.8  |
> | AR2-G (This work)                     | Base  | 58.7 | 77.9 | 86.0 | 90.1  |
> | AR2-G (This work)                     | Large | 61.1 | 78.8 | 86.5 | 90.4  |

---

### Decision · Program_Chairs · 2022-01-20

**Decision:**

Accept (Poster)

**Comment:**

This paper introduces a new method for jointly training a dense bi-encoder retriever with a cross-encoder ranker. More precisely, the proposed method is iteratively training the retriever and the ranker, using an objective function inspired by adversarial training. In addition, the authors propose to use a distillation loss from the ranker to the retriever as a regularization term. The proposed method, called AR2, is evaluated on three retrieval benchmarks from question answering: NaturalQuestions, TriviaQA and MS-MARCO. The method obtains state-of-the-art retrieval performance on these three datasets.

Overall, the reviewers agree that the strong performance obtained by the proposed method is a strength of the paper. Regarding novelty, some reviewers argue that the method is a combination of existing techniques, hence lacking novelty, while the others believe that combining these different techniques is novel enough. Regarding the experimental section, some concerns were raised about comparisons with previous work (eg, BERT vs ERNIE) or the fact that it was a bit hard to determine where the improvements come from. I believe that these concerns were well addressed by the authors, and I tend to believe that combining existing techniques to obtain a strong system is novel enough. I thus lean towards accepting this paper to the ICLR conference.